# Reversal of Peripheral Neuropathic Pain by the Small-Molecule Natural Product Narirutin via Block of Na_v_1.7 Voltage-Gated Sodium Channel

**DOI:** 10.3390/ijms232314842

**Published:** 2022-11-27

**Authors:** Haoyi Yang, Zhiming Shan, Weijie Guo, Yuwei Wang, Shuxian Cai, Fuyi Li, Qiaojie Huang, Jessica Aijia Liu, Chi Wai Cheung, Song Cai

**Affiliations:** 1Department of Anatomy and Histology, Shenzhen University Health Science Center, Shenzhen 518060, China; 2Laboratory and Clinical Research Institute for Pain, Department of Anaesthesiology, School of Clinical Medicine, Li Ka Shing Faculty of Medicine, The University of Hong Kong, Hong Kong 999077, China; 3Department of Anesthesiology, Shenzhen People’s Hospital (The First Affiliated Hospital, Southern University of Science and Technology; The Second Clinical Medical College, Jinan University), Shenzhen 518020, China; 4Shenzhen Engineering Research Center of Anesthesiology, Shenzhen 518020, China; 5Department of Neuroscience, City University of Hong Kong, Hong Kong 999077, China

**Keywords:** natural products, Narirutin, Na_v_1.7, ion channels, nociceptive neurons, neuropathic pain

## Abstract

Neuropathic pain is a refractory chronic disease affecting millions of people worldwide. Given that present painkillers have poor efficacy or severe side effects, developing novel analgesics is badly needed. The multiplex structure of active ingredients isolated from natural products provides a new source for phytochemical compound synthesis. Here, we identified a natural product, Narirutin, a flavonoid compound isolated from the *Citrus unshiu*, showing antinociceptive effects in rodent models of neuropathic pain. Using calcium imaging, whole-cell electrophysiology, western blotting, and immunofluorescence, we uncovered a molecular target for Narirutin’s antinociceptive actions. We found that Narirutin (i) inhibits Veratridine-triggered nociceptor activities in L4-L6 rat dorsal root ganglion (DRG) neurons, (ii) blocks voltage-gated sodium (Na_V_) channels subtype 1.7 in both small-diameter DRG nociceptive neurons and human embryonic kidney (HEK) 293 cell line, (iii) does not affect tetrodotoxin-resistant (TTX-R) Na_V_ channels, and (iv) blunts the upregulation of Na_v_1.7 in calcitonin gene-related peptide (CGRP)-labeled DRG sensory neurons after spared nerve injury (SNI) surgery. Identifying Na_v_1.7 as a molecular target of Narirutin may further clarify the analgesic mechanism of natural flavonoid compounds and provide an optimal idea to produce novel selective and efficient analgesic drugs.

## 1. Introduction

Neuropathic pain is a growing global affliction affecting millions of people’s lives [1]. Alleviating neuropathic pain is challenging, as present clinical, widely used analgesics either have poor curative effects or are restricted by severe adverse effects [2,3]. There is thus a vast unmet need for the development of novel analgesics that effectively inhibit nociceptors without off-target effects on motor or central neurons. 

In the quest for new analgesics, natural products and their derivates remain interesting therapeutic resources. The multiple structures of active components isolated from natural products provide new templates for developing novel analgesics [4,5]. Unlike artificial synthetic small-molecule compounds, natural products usually exhibit better biocompatibility and fewer side effects [6]. However, one difficulty of natural products in clinical translation is identifying their targets and underlying mechanisms to maximize their efficacy while reducing side effects.

The DRG, which is involved in the transduction of pain signaling from the peripheral nervous system to the central nervous system, is an attractive therapeutic target for pain alleviation [7]. Na_V_ channels, especially subtype Na_v_1.7 and Na_v_1.8, play a fundamental role in the initiation and propagation of action potentials in DRG electrically excitable neurons, making them potential targets for the development of novel analgesics [8]. Recent in vivo and in vitro studies have identified inhibiting Na_V_ channels as a mechanism of natural products in pain therapy [9,10]. For instance, protoxin-II, a peptide from *tarantula thrixopelma pruriens*, blocks the Na_v_1.7 channel at picomolar concentrations [11]. Loperamide, known initially as an anti-diarrheal drug, was identified as a Na_v_1.7 blocker that displayed a mild use- and state-dependent inhibition on Na_v_1.7 channels [12]. 

Among these natural products, the flavonoid is a series of compounds that have been thoroughly investigated in pain therapy (catechin [13], pinostrobin [14], quercetin [15], and naringenin [16]). Narirutin is one of the flavonoid active constituents isolated from the *Citrus unshiu*. Narirutin has been reported to (i) inhibit malignant cancer cell growth and angiogenesis via the PI3K/AKT and ERK/MAPK signal transduction cascades [17], (ii) have neuroprotective properties via discharging nitrosative and oxidative stress through reactive oxygen species and activating voltage-gated potassium channels [18], (iii) suppress inflammation observed in mice macrophage cell lines [19], and (iv) present antioxidative and hepato-protective properties in normal rat liver cells [20]. In addition to these bioactivate effects, one of Narirutin’s analogues, naringenin, was reported to inhibit mechanical allodynia in acute and chronic rodent pain models [16]. These reports inspired us to hypothesize a possible analgesic effect that Narirutin may possess and to seek its possible mechanism.

In this study, we identified Narirutin as an efficient analgesic in alleviating neuropathic pain in rodent models of spared nerve injury. We performed in vitro experiments in both DRG neurons and HEK293 cell lines to identify Na_v_1.7 as a molecular target of Narirutin, which may promote flavonoids as a structural template for novel analgesics development.

## 2. Results

### 2.1. Narirutin Shows Antinociceptive Effects in SNI-Induced Rat Models of Neuropathic Pain

Narirutin (C_27_H_32_O_14_, PubChem CID 442431, molecular weight 580.5 g/mol; Figure 1a) is a flavonoid with reported antioxidative and anti-inflammatory actions [21]. As a disaccharide derivative of naringenin, whether Narirutin has the same analgesic effect as naringenin [16] has not been confirmed. Here, we set out to evaluate its analgesic effect and investigate its possible mechanism of action. To assess Narirutin’s potential antinociceptive activity, we selected the SNI neuropathic pain model. The model involves a lesion of two of the three terminal branches of the sciatic nerve (tibial and common peroneal nerves), keeping the sural nerve intact, and could result in early and prolonged peripheral neuropathic pain [22]. Six adult male rats were subjected to SNI surgery (see Methods), and the paw withdrawal threshold was measured 14 days post-injury (dpi). In this allodynic state, the rats were intrathecally treated with vehicle (10% DMSO, 10% Tween-80, 80% Saline, 20 μL) or Narirutin (20 μg/20 μL). Paw withdrawal thresholds in SNI rats treated with Narirutin were significantly increased at 1 h post-treatment, lasting for about 3 h (Figure 1b). An increased area under the curve supported these data as a result of treatment with Narirutin compared to vehicle rats (Figure 1c). Moreover, we also performed acetone test to evaluate Narirutin’s antinociceptive capacity on cold allodynia. After treatment with Narirutin, cold allodynia was significantly decreased (Figure 1d,e). Together, these data demonstrate the antinociceptive potential of Narirutin in nerve injury-induced neuropathic states.

### 2.2. Narirutin Inhibited Veratridine-Triggered Nociceptor Activities of DRG Neurons

We focused our efforts on voltage-gated sodium channels, which are highly expressed on afferent pain fibers and are involved in transmitting pain signals [23,24,25]. As a potent Voltage-gated sodium channel activator, Veratridine (VTD) triggers Na^+^ influx to elicit action potential and prompts Ca^2+^ entry into neurons, while KCL can directly induce action potential through membrane depolarization. Thus, to assess Narirutin’s potential in modulating the Na_V_ channel’s function, we used a 30 µM VTD or 40 mM KCL stimulus to open Na_V_ and Ca_V_ channels in DRG sensory neurons, thus permitting measurements of evoked Ca^2+^ influx. In the results, Narirutin inhibited VTD-triggered Ca^2+^ activities in DRG sensory neurons (Figure 2a up, DMSO = 1.4 ± 0.0022, *n* = 512; Narirutin = 1.2 ± 0.0018, *n* = 558, *p* = 0.0260). However, no inhibition could be found on KCL-triggered Ca^2+^ activity after treatment with Narirutin (Figure 2a down). As VTD specifically functions on the Na_V_ channel, these data imply that Narirutin affects Na_V_ rather than Ca_V_ channels. Given that nociceptors and non-nociceptors respond to VTD with distinct response profiles [26], a VTD-based neuron classification assay helps us to better investigate the potential analgesic mechanism of Narirutin. Generally, VTD elicits four distinct response profiles in sensory neurons: oscillatory (OS), slow decay (SD), rapid decay (RD), and intermediate decay (ID) (Figure 2b). VTD elicits OS on nociceptors, while most non-nociceptors showed SD [26,27]. In this study, Narirutin exhibited a ~26% decrease in the population of OS neurons (DMSO = 39.2 ± 4.1%; Narirutin = 28.9 ± 5.7%, *p* = 0.0189), while the population of SD (DMSO = 20.1 ± 1.1%; Narirutin = 18.0 ± 1.3%, *p* = 0.9896) and Veratridine (DMSO = 29.6 ± 1.1%; Narirutin = 32.3 ± 1.6%, *p* = 0.9333) neurons remained unchanged (Figure 2c). These results indicate that Narirutin relieves mechanical allodynia of SNI rats by suppressing nociceptors activities.

### 2.3. Total Sodium Current (INa) in DRG Sensory Neurons Is Reduced by Narirutin Treatment

As sodium ions are critical components in generating action potentials and modulating neuronal excitability and propagating nociceptive signaling, we used whole-cell voltage-clamp electrophysiology to assess the effects of Narirutin on Na_V_ currents in DRG nociceptive neurons. First, total Na_V_ currents were measured from rat DRG small-diameter (<30 μm) neurons. Typical families of total sodium currents from DRG neurons treated with 0.1% DMSO or a 20 μM concentration of Narirutin overnight are shown in Figure 3a. To do so, we used a current-voltage protocol, holding rat DRG sensory neurons at −80 mV with depolarization steps (200 ms step) from −70 mV to +60 mV in 5 mV increments (Figure 3b). When compared with the control (0.1% DMSO), Narirutin inhibited total sodium current density with a ~48.59% decrease in peak current density (Figure 3b,c, 0.1% DMSO, −931.7 ± 76.3 (pA/pF) (*n* = 9); Narirutin, −504.2 ± 79.9 (pA/pF) (*n* = 10), *p* = 0.0016; Mann–Whitney test). The data were normalized according to cell capacitance to account for the heterogeneity of DRG neuronal populations. To rule out changes in channel gating as the cause of the inhibitory properties of Narirutin on total sodium currents, we investigated the effect of Narirutin on the biophysical properties of activation and inactivation of the DRG sodium currents, as described in the Methods. After converting the current values to conductance (G), the conductance-voltage relationship was fitted with a Boltzmann equation. The G value for each neuron was normalized to the maximal value (G_max_) derived from the fit. The Boltzmann relevant factors, similar to half-maximal activation (V_1/2_) and slope (*k*), for the single fits to the data, are shown in Figure 3d and summarized in Table 1. Similarly, steady-state inactivation properties of sodium currents were compared between the conditions (Figure 3e). These results show that Narirutin shifted both the V_1/2_ activation (left-shifted) and inactivation (right-shifted) of Na_V_ channels (see Table 1).

### 2.4. Narirutin Inhibits TTX-Sensitive Rather Than TTX-Resistant Sodium Currents in DRG Sensory Neurons

To delineate the specific type of sodium channel affected, we assessed the possible effect of Narirutin on TTX-S (Na_v_1.1–1.7) and TTX-R (Na_v_1.8, 1.9) Na^+^ channels. First, we used a post hoc subtraction method [28] to isolate the effect of Narirutin on TTX-S Na^+^ channels, which activate at low thresholds and are fast-inactivating. Narirutin significantly inhibited TTX-S Na^+^ currents (Figure 3f, 0.1% DMSO, −553.9 ± 34.0 pA/pF (*n* = 9); Narirutin, −389.7 ± 41.9 pA/pF (*n* = 10), *p* = 0.0097; Mann–Whitney test). Then, after incubating overnight with Narirutin 20 μM or 0.1% DMSO, two groups of DRG neurons were cotreated with TTX 1 μM acutely, and then TTX-R Na^+^ currents were recorded (Figure 4a). As a result, Narirutin did not significantly inhibit TTX-R Na^+^ currents (Figure 4b,c, 0.1% DMSO, −486.4 ± 49.8 pA/pF (*n* = 17); Narirutin, −448.0 ± 98.8 pA/pF (*n* = 10), *p* = 0.6544; Mann–Whitney test). We then explored whether the voltage-dependent activation and inactivation properties of TTX-R Na^+^ channels in DRG neurons were affected by Narirutin by comparing the midpoint of whole-cell ionic conductance (V_1/2_) and the slope factor (k) at the command voltage. Data with representative Boltzmann fits for either DMSO or Narirutin treatment are shown in Figure 4d,e and summarized in Table 1. The activation properties of TTX-R Na^+^ currents were right-shifted, while inactivation properties remained unchanged after treatment with either condition. These results show that Narirutin inhibits TTX-S rather than TTX-R Na^+^ currents in DRG sensory neurons (Table 1).

The Na_v_1.8 channel is one of the TTX-R channels highly correlated with nociceptive signal transmission [29,30]. To corroborate the above findings pharmacologically, we used the specific Na_v_1.8 inhibitor A-803467. DRG neurons were cotreated with 500 nM A-803467 acutely after overnight 0.1% DMSO or 20 μM Narirutin administration. After excluding Na_v_1.8 by A-803467, Narirutin can still significantly inhibit the rest of the Na_V_ subtype channels with a 41.72% decrease in peak current density (Figure 5b,c, 0.1% DMSO, −798.1 ± 49.1 pA/pF (*n* = 10); Narirutin, −465.2 ± 103.5 pA/pF (*n* = 8), *p* = 0.0044; Mann–Whitney test) as well as a right shift of the activation properties (Figure 5d,e, Table 1). These results show that Narirutin specifically inhibits TTX-S rather than TTX-R Na^+^ currents.

### 2.5. Narirutin Specifically Inhibits Na_v_1.7 Currents in Both DRG Sensory Neurons and HEK293 Cell Lines

Since Narirutin can inhibit TTX-S rather than TTX-R sodium currents, we wondered if it can specifically target a single subtype of TTX-S Na^+^ channel. In DRG neurons, TTX-S currents are fast-inactivating, primarily carried by Na_v_1.1, 1.3, 1.6, and 1.7 in DRG, and shape the action potential and consequent requirement for initial depolarization. Among these subtype channels, Na_v_1.7 functions as a threshold channel to generate and propagate action potentials and modulate neuron excitability [31]. Therefore, we used a specific Na_v_1.7 inhibitor, PF-05089771, to clarify if Narirutin can specifically target on the Na_v_1.7 channel. DRG neurons were cotreated with 100 nM PF-05089771 acutely after incubating 0.1% DMSO or 20 μM Narirutin overnight. In DRG neurons treated with both compounds, the inhibition effect of Narirutin on total sodium current is blunted by PF-05089771 in DRG sensory neurons (Figure 6b,c, 0.1% DMSO, −590.1 ± 71.2 pA/pF (*n* = 12); Narirutin, −497.7 ± 43.0 pA/pF (*n* = 11), *p* = 0.5254; Mann–Whitney test). Activation properties between the two groups show no significant difference (Figure 6e,f, Table 1).

To further confirm that Na_v_1.7 is the particular target of Narirutin, we evaluated Narirutin’s direct effect on the Na_v_1.7 channel in HEK293 cell lines that were transfected using pcDNA3.1- Na_v_1.7-Flag with GFP plasmid and incubated overnight with 0.1% DMSO or Narirutin 20 μM for whole cell patch recording. Compared with the control (0.1% DMSO), Narirutin treatment resulted in a 52.78% decrease in peak current density of Na_v_1.7 (Figure 7b,c, 0.1% DMSO, −178.7 ± 25.7 pA/pF (*n* = 23); Narirutin, −84.4 ± 19.4 pA/pF (*n* = 10), *p* = 0.0163; Mann–Whitney test), as well as a right shift of the activation properties (Figure 7d,e, Table 1). These results indicate that inhibition of TTX-S currents with Narirutin is due to Na_v_1.7.

### 2.6. Narirutin Blunts the Upregulation of Na_v_1.7 in CGRP+ DRG Sensory Neurons after SNI Surgery

Since Narirutin can selectively inhibit Na_v_1.7 channels, we hypothesized that in addition to direct inhibiting, Narirutin might affect the expression level of Na_v_1.7 on DRG nociceptive neurons. To confirm this hypothesis, we investigated the effect of Narirutin on Na_v_1.7 expression in L4-L6 DRGs of SNI rats. However, Western Blotting did not show any profound alterations in the total protein expression of Na_v_1.7 among three groups of DRGs: sham, SNI, and SNI treated with Narirutin (Figure 8a,b). As the calcium imaging data indicate that Narirutin mainly affects nociception neurons, we then co-immunostained Na_v_1.7 with CGRP, IB4, and NF200 separately to identify the effects of Narirutin on the expression level of Na_v_1.7 on different types of neurons. Interestingly, we found that the percentage of Na_v_1.7-positive cells in CGRP-labeled neurons is increased, and such increment can be blunted by Narirutin administration (Figure 8c,d). However, such an effect cannot be found in IB4- or NF200-labeled neurons. These results indicate that Narirutin could downregulate the expression of Na_v_1.7 on CGRP+ DRG nociceptive neurons.

## 3. Discussion

The two salient findings of our work are: (i) the small-molecule natural product Narirutin, derived from *Citrus unshiu*, has antinociceptive potential in neuropathic pain, and (ii) Narirutin targets voltage-gated sodium subtype 1.7 (Na_v_1.7) channels to achieve this effect. We arrived at this mechanistic specificity conclusion by noting (i) the lack of effect of Narirutin on tetrodotoxin-resistant channels like Na_v_1.8 in sensory neurons and (ii) the selective inhibition effect of Narirutin on Na_v_1.7 channels in DRG sensory neurons and the HEK293 cell line. Equally importantly, our data demonstrate that the underlying mechanism may be involved with downregulating the expression level of Na_v_1.7 on DRG CGRP+ nociceptive neurons.

The effects of analgesics on nociceptors and non-nociceptors are different. People seek ideal analgesics that have a specific effect on nociceptors, with minimal or no effect on non-nociceptors (as a proxy for unwanted adverse effects on CNS and motor neurons). To achieve this goal, Mohammed and colleagues developed a simultaneous assessment of drugs’ action on nociceptors and non-nociceptors based on neurons’ distinct response profiles to VTD. They, therefore, provided an efficient and more informative assay for screening for analgesics on sensory neurons [26]. Generally, VTD elicits four distinct response profiles in different sensory neurons: oscillatory (OS), slow decay (SD), rapid decay (RD), and intermediate decay (ID). OS stands for nociceptors, while most non-nociceptors showed SD. Changes in the OS population reflect changes in nociceptors. Changes in the VTD-irresponsive population (VTD-) can reflect the sensitization of high-threshold and usually “silent” neurons. The two minor profiles, ID and RD, are mostly nociceptors but can be excluded from an assay for simplicity as both account for less than 5% to 10% of all neurons. In this study, we conducted a calcium imaging assay according to the above protocol to evaluate the effect of Narirutin on different types of neurons. First, we evaluated the peak calcium responses of DRG neurons induced by VTD after Narirutin incubation. As a result, Narirutin inhibits VTD-triggered Ca^2+^ activities in DRG neurons. Then, after dividing these neurons into four classes, we identified Narirutin as an ideal analgesic based on its ability to reduce the OS population without affecting the SD and VTD-populations.

Since the function of Na_v_1.7 is highly correlated with the OS population [26], we then investigated if Na_v_1.7 is the target of Narirutin. By using whole-cell voltage-clamp recording, we not only evaluated the effect of Narirutin on total sodium currents, but also clarified its function in subtypes of Na_v_1.7 currents by isolating channels electrically and pharmacologically in DRG small-diameter sensory neurons, which theoretically consist of nociceptive neurons that belong to the OS population. Electrically, The H-infinity protocol (see Methods) allowed subtraction of electrically isolated TTX-R (current available after −40 mV prepulse) from the total current (current available after −120 mV prepulse) to estimate the TTX-S current. This protocol is possible because of the differential inactivation kinetics of TTX-R vs. TTX-S channels, wherein only the TTX-R current becomes activated and TTX-S is fast inactivated at −40 mV [28]. Pharmacologically, TTX and two selective inhibitors (PF-05089771 for Na_v_1.7 and A-803467 for Na_v_1.7) are used for subtype Na_V_ currents isolation. TTX and A-803467 helped us double confirm that Narirutin mainly affects TTX-S currents. PF-05089771 is an aryl sulfonamide Na_v_1.7 channel blocker that binds to the inactivated state of Na_v_1.7 channels with high affinity [32]. This property allowed us to isolate Na_v_1.7 from total sodium currents and conclude that Na_v_1.7 is a possible target of Narirutin. It is worth noting that, although PF-05089771 inhibits Na_v_1.7, it can exert additional confounding pharmacologic activities. Therefore, we evaluated Narirutin’s direct effect on the Na_v_1.7 channel in the HEK 293 cell line, which pcDNA3.1- Na_v_1.7-Flag transfected, to further confirm that Na_v_1.7 is the certain target of Narirutin.

Highly expressed Na_v_1.7 in DRG neurons is associated with pain progress [33,34,35]. However, different from paclitaxel-induced neuropathic pain rodent models [36], both a previous study [37] and our results demonstrate that no significant difference could be found in the total expression level of Na_v_1.7 on ipsilateral L4-6 DRGs between sham and SNI rats. This distinction impels us to explore its underlying reason. Inspired by our calcium imaging results indicating that Narirutin presents disparate effects on four types of neurons, we hypothesis that Na_v_1.7 expression in different types of neurons after SNI is varied. Since the small-diameter neurons are mainly C-fiber-related nociceptors [38,39], we conducted immunostaining CGRP and IB4 for labeling unmyelinated peptidergic or non-peptidergic sensory neurons, and NF200 for myelinated neurons [40]. Calcitonin gene-related peptide (CGRP), a 37 amino-acid neuropeptide found mostly in peptidergic sensory C-fibers, has been suggested to be implicated in the pathogenesis of neuropathic pain [41], which is likely mediated by modulating nociception and sustaining neurogenic inflammation that leads to further peripheral and central pain sensitization [42]. As expected in our study, Na_v_1.7 expression is varied among three types of neurons after SNI, as an increment is shown in CGRP-labeled neurons, and such increment can be blunted by Narirutin administration. These findings may indicate a possible mechanism that Narirutin may affect through downregulating the expression of Na_v_1.7 in DRG CGRP+ peptidergic nociceptive neurons. Given CGRP+ peptidergic nociceptive neurons are proved to be essential for tissue regeneration [43], it is worth studying in the future whether Narirutin can also serve as a novel drug in nerve regeneration. 

Several limits exist in this study. First, we did not investigate Narirutin’s effect on Ca^2+^ or K^+^ channels, two other voltage-gated channels highly correlated with nociceptive signaling transmission. Second, although we concluded from whole-cell patch recording that Narirutin can selectively inhibit Na_v_1.7 channels, how Narirutin directly or indirectly acts on Na_v_1.7 remains unclear. Third and last, given the different reaction to drugs among rodents and humans, clinical trials or in vitro human-derived DRG neuron experiments are needed to further verify Narirutin’s efficacy in pain therapy.

In our study, we add to the evidence supporting Narirutin’s pain-relieving potential with data from SNI models of neuropathic pain, thus supporting its antinociceptive property and relevance to neuropathic pain treatment from nerve injury. Identifying the Na_v_1.7 channel as a target of Narirutin’s activity could further clarify the analgesic mechanism of flavonoid natural compounds and provide an optimal path for producing novel selective and efficient analgesic drugs.

## 4. Materials and Methods

### 4.1. Animals

The animals required for the experiments were obtained from Beijing Viton Lever. Animals (SD, Male rats) were housed in a pathogen-free environment with controlled conditions of temperature (23 ± 3 °C) and light (12 h light/12 h dark cycle; lights on from 7:00–19:00); water and standard rodent food were freely available. The Animal Care and Use Committee of the Shenzhen University Medical School approved all experiments. All experiments were conducted in compliance with the Guide for the Care and Use of Laboratory Animals and the ethical guidelines of the International Association for the Study of Pain. In the behavioral experiments, animals were randomly assigned to the treatment or control groups. All behavioral experiments were performed by experimenters blinded to the experimental group and the treatment method.

### 4.2. Reagents

Unless otherwise stated, reagents were obtained from Sigma-Aldrich Chemicals (St. Louis, MO, USA).

Neutral protease (Worthington, LS02104), IA collagenase (Worthington, LS004194), Poly-D-lysine hydrobromide (Sigma, P7886), DMEM (Gibco, 2318815), fetal bovine serum (Gibco, 2176404), penicillin/streptomycin (Gibco, 2321127), JetPrime (Polyplus, 101000046), TTX (Alomone, T-550), A803467 (Alomone, A-105), PF-05089771 (Alomone, P-315).

### 4.3. Spared Nerve Injury (SNI)

The SNI model was constructed as described previously in [22]. The model was constructed using isoflurane anesthesia in rats, exposing the sciatic nerve to the left hind leg’s common peroneal, tibial, and peroneal nerves, then ligating and clipping the first two nerves and preserving the peroneal nerve. In the sham-operated group, only the sciatic nerve was exposed and no ligation or injury was done.

### 4.4. Assessment of Mechanical Allodynia Using the Von Frey Test

The animals were placed in plastic cages with metal mesh on the bottom and allowed to acclimatize for at least one hour before each test. Each Von Frey wire was held perpendicular to the contact surface of the paw until it bent. Tests were carried out and data collected using Dixon’s “up and down” method (successive increases and decreases in stimulus intensity) [44]. 

### 4.5. Assessment of Cold Allodynia Using the Acetone Test

The animals were placed in plastic cages with metal mesh on the bottom and allowed to acclimatize for at least one hour before each test. A quantity of 0.15 mL of acetone was squeezed out of a 1 mL syringe with the needle removed and smeared onto the soles of the rats, allowing the acetone to diffuse and evaporate in their plantar skin, thereby producing a cooling effect. Five experiments were performed for each rat, each more than 2 min apart (to avoid the effects of residual acetone). The presence of noxious withdrawal responses (e.g., foot tremors, leg shaking in the air, foot licking, etc.) was recorded in the rats during the five experiments [45].

### 4.6. Cell Culture and Transfection

#### 4.6.1. Cell Culture

DRG neurons were obtained from SD rats by acute isolation. SD rats were euthanized at 4–6 weeks (anesthetized with isoflurane and decapitated). Remove 20–25 spinal dorsal root neurons in the intervertebral foramen. The collected DRG were digested in a mixture of enzymes (neutral protease and type IA collagenase). Incubate in a shaker at 37 °C for about 40 min, centrifuge (27 °C, 800 r/min) for 3 min, discard the supernatant and leave to sink. DRG neurons were resuspended in mixed medium (90% DMEM, 10% fetal bovine serum, 1% penicillin/streptomycin). Subsequently, DRG neurons were inoculated on glass crawl sheets previously treated with poly-d-lysine (50 μg/mL). DRG neurons were cultured in the mixed medium and used within 48 h.

HEK293 cells were purchased from Shanghai Institute of Biological Sciences, Chinese Academy of Sciences (Shanghai, China). HEK293 cells were cultured in medium (90% DMEM, 10% fetal bovine serum, 1% penicillin/streptomycin) in an atmosphere of 5% CO_2_ at 37 °C.

#### 4.6.2. Transfection

The Mouse Na_v_1.7 sequence was synthesized by GenScript and inserted into the pcDNA3.1(+)-C-DYK plasmid between the BamH1 and EcoRI restriction sites. This construct allows for the expression of Na_v_1.7 under a cytomegalovirus promoter (CMV) and with an intracellular Flag tag located at the C-terminus of the protein. The plasmid of GFP was purchased from Shenzhen Yanming Biotechnology Company. HEK293 cells were transiently transfected with JetPrime transfection reagent according to the manufacturer’s instructions. HEK293 cells were transfected for 6–8 h using pcDNA3.1-Na1.7-Flag in conjunction with the GFP plasmid. Subsequently, the medium was replaced, compounds were added and incubated overnight for whole-cell patch recording.

### 4.7. Calcium Imaging and Analysis

DRG neurons were incubated for 30 min at 37 °C with 3 μM Fura-4AM (Thermo Fisher, # F14201, stock solution prepared at 1 mM in DMSO) to explore changes in intracellular Ca^2+^. During the experiments, the cultured neurons was continuously perfused with a standard bath solution (mM) containing 139 NaCl, 3 KCl, 0.8 MgCl_2_, 1.8 CaCl_2_, 10 HEPES, 5 glucose, pH 7.4 with NaOH. Veratridine (MCE, #HY-N6691, 30 µM) was added after 1 min stable baseline (F0) recording for 3 min and then switched back to standard bath. Ringers with 40 mM KCl was perfused at the end of recordings to identify viable neurons. The perfusion was stopped and the changes of intracellular Ca^2+^ concentration were examined with a ratio of ∆F/F0, calculated after subtracting background. Neurons were included in the analysis if they responded to 40 mM KCl. On rare occasions, neurons responded to VTD but not KCl (or the KCl response was not clear due to the Ca^2+^ signal not returning to baseline after the application of last agonist). We defined a response as an increase in (∆F/F0) ratio of 2 SD above the baseline. Mean values from this experiment were compared with each other by T-test. Fluorescence imaging was performed with an inverted microscope, Nikon Eclipse Ti-U (Nikon Instruments Inc), using objective Nikon Plan Fluor 4× 0.13 and a Photometrics cooled CCD camera CoolSNAP HQ2 controlled by NIS Elements software (Nikon instruments, version AR 5.30.03). The excitation light was excited by current regulated power supply (HAMAATSU PHOTONICS KK) at 488 nm and delivered by a Lambda 10-3 system (Sutter Instruments). To minimize photobleaching and phototoxicity, the images were taken every 5 s. During the time-course of the experiment, the minimal exposure time was used to acquire acceptable image quality.

### 4.8. Whole-Cell Voltage Clamp Electrophysiology

The composition of the extracellular recording solution used to record sodium currents was (mM): 140 NaCl, 30 tetraethylammonium chloride, 10 D-glucose, 3 KCl, 1 CaCl_2_, 0.5 CdCl_2_, 1 MgCl_2_, and 10 HEPES, and the pH of the solution was adjusted with NaOH to 7.3 with an osmolarity of 310–315 mOsm/L. The composition of the intracellular recording solution was (mM): 140 CsF, 10 NaCl, 15 HEPES and 1.1 Cs-EGTA, and the pH of the solution was adjusted to 7.3 using CsOH. All recordings were obtained in acutely isolated DRG neurons from SD rats. DRG neurons were recorded using current-voltage (I-V) and activation/inactivation voltage protocols. The voltage protocols were as follows: (i) I-V protocol: DRG neurons were clamped at −80 mV, and the cells were depolarized in 200 ms steps in the range of −70 to +60 mV (+5 mV increments). This was done to obtain a current density such that the activation of sodium channels was generated between approximately 0–10 mV. This process occurs between 0 and 10 mV and can be analyzed as a function of voltage, as inferred from the peak current density (normalized to the cell capacitance (picofarad, pF)); (ii) Inactivation protocol: Starting from a clamp potential of −60 mV, a hyperpolarization/depolarization pulse of 1 s in the range of −120 to 20 mV, in steps of +10 mV for 1 s. This incremental increase in membrane potential puts a different proportion of sodium channels into a state of rapid deactivation—in this case, a test pulse of 0-mV for 200 ms shows rapid deactivation when normalized to the maximum sodium current. TTX-S was isolated using 1 µM of TTX [28], Na_v_1.8 was isolated using 500 nM of selective blocker A-803467 [46], and Na_v_1.7 was isolated using 100 nM of PF-05089771 [47]. Because of the differential inactivation kinetics of TTX-resistant and TTX-sensitive channels, the fast inactivation protocol allowed subtraction of electrically isolated TTX-R (current available after −40 mV prepulse) from total current (current available after −120 mV prepulse). 

For the Na_v_1.7 experiment we used HEK293 cells that predominantly expressed Na_v_1.7 after transient transfection. Electrophysiological recording methods are as described in the previous section.

Whole-cell recording was performed using a HEKA EPC-USB 10 (HEKA Instruments). Data were obtained by Patchmaster (HEKA) and analysed with Fitmaster (HEKA). Capacitive artifacts were fully compensated, and series resistance was compensated by ~60%. The pipettes used had a resistance of 1–4 MΩ. All experiments were performed at room temperature.

### 4.9. Immunoblotting

Western blotting method. L4-6 DRG neurons were extracted from the ipsilateral surgery of Sham or SNI rats and stored in liquid nitrogen to be set aside. L4-6 DRG neurons were sheared in pre-cooled potent RIPA lysate (containing protease and phosphatase inhibitors) and their whole cell lysates were obtained by ultrasonic crushing on ice. Whole-cell protein extracts were separated using 8% SDS-PAGE gels and transferred to PVDF membranes (activated with methanol). Protein blots were closed in 5% skimmed milk (Tris buffer configuration containing 0.1% Tween-20) for 1 h, then incubated overnight with SCN9A antibody (Alomone, ASC-008, 1:1000) or GAPDH antibody (Cell Signaling, 5174, 1:1000), respectively, and the next day for 1 h with the corresponding secondary antibody. Immunoreactive bands were detected by chemiluminescence. 

### 4.10. Immunofluorescence

Frozen sections. Rats were transcardially perfused with 4% PFA and the left side of L4-L6 was subsequently removed and fixed in 4% PFA for 1 h and subsequently transferred to a 30% sucrose solution for over 48 h for dehydration. The DRG was divided into 7 µm frozen sections using a freezing microtome (Leica, Nussloch, CM1950). DRG sections were washed with PBS and blocked with donkey serum for 1 h before being incubated with SCN9A (Alomone, ASC-008, 1:200), CGRP (Abcam, ab81887, 1:200), or NF200 (Sigma, N0142, 1:200) for more than 14 h at 4 °C. After washing the primary antibody, the slices were incubated for 1 h at room temperature with Cy3 coupled antibody (Jackson ImmunoResearch, 1:400), FITC secondary antibody (Jackson ImmunoResearch, 1:400), or IB4 (Sigma, L2895, 1:50); washed; and then blocked with anti-fluorescence quencher containing DAPI. Fluorescent images of DRG were acquired using laser confocal microscopy (Leica, SP8).

Images of the DRG were analyzed using ImageJ (3 slices per rats, *n* = 3 per group). The number of positive neurons in DRG sensory neurons labelled using CGRP, IB4 or NF200 was recorded and co-localization with Na_v_1.7 and specific neuronal markers (yellow) was recorded. The percentage expression of Na_v_1.7 in different neurons was calculated and analyzed using Prism 9.

### 4.11. Data Analysis

Von Frey and acetone test data were analyzed using Excel (Microsoft), and prism 9. Data are presented as mean ± SEM. The two-way ANOVA determined statistical significance with Sidak’s post hoc test. Calcium imaging data were analyzed using Excel (Microsoft), and Graphpad prism 9. Statistical significance was determined by the unpaired two-tailed Student’s *t* test and two-way ANOVA with Sidak’s post hoc test. Whole-cell voltage clamp data were analyzed using Fitmaster (HEKA Electronics), Excel (Microsoft), and prism 9. Statistical significance was determined by Mann–Whitney test. Immunofluorescence co-localization was analyzed using Excel and Prism 9, and Statistical significance was determined by one-way ANOVA followed by Tukey’s test. All data are expressed as mean ± standard error of the mean (SEM). Results were judged to be significant when *p* < 0.05.

## Figures and Tables

**Figure 1 ijms-23-14842-f001:**
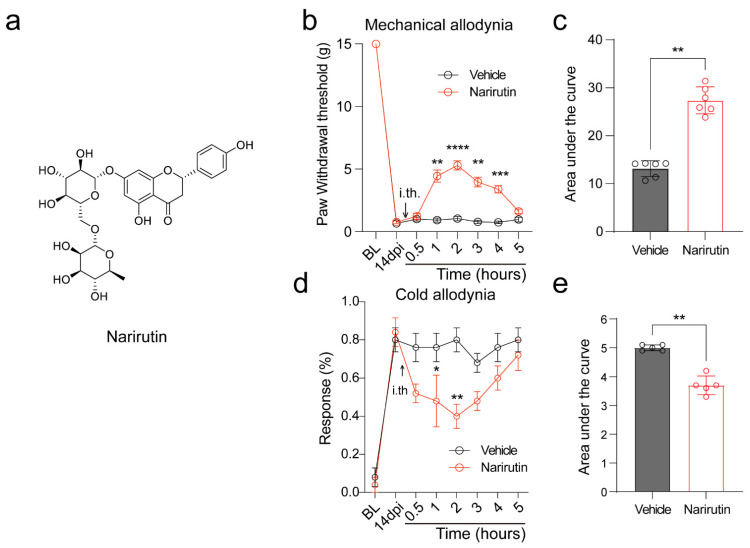
Narirutin alleviates SNI-induced mechanical allodynia. (**a**) Chemical structure of Narirutin. (National Center for Biotechnology Information (2022). PubChem Compound Summary for CID 442431, Narirutin. Retrieved 22 September 2022, from https://pubchem.ncbi.nlm.nih.gov/compound/Narirutin.) (**b**) The Paw withdrawal threshold (PWT) and (**d**) acetone noxious withdrawal responses of adult male rats (*n* = 6) were measured at 14 dpi. Rats were treated intrathecally (i.th., black arrow) via a catheter with vehicle (10% DMSO, 10% Tween-80, 80% Saline, 20 μL) or Narirutin (20 μg/20 μL) as indicated. Area under the curve for (**c**) mechanical and (**e**) cold allodynia was derived again as indicated before using Graphpad Prism. Data are expressed as means ± SEM. Asterisks indicate statistical significance compared with vehicle treatment (* *p* < 0.05, ** *p* < 0.01, *** *p* < 0.001, and **** *p* < 0.0001; two-way ANOVA with Sidak’s post hoc test). The experimenter was blinded to the treatment condition.

**Figure 2 ijms-23-14842-f002:**
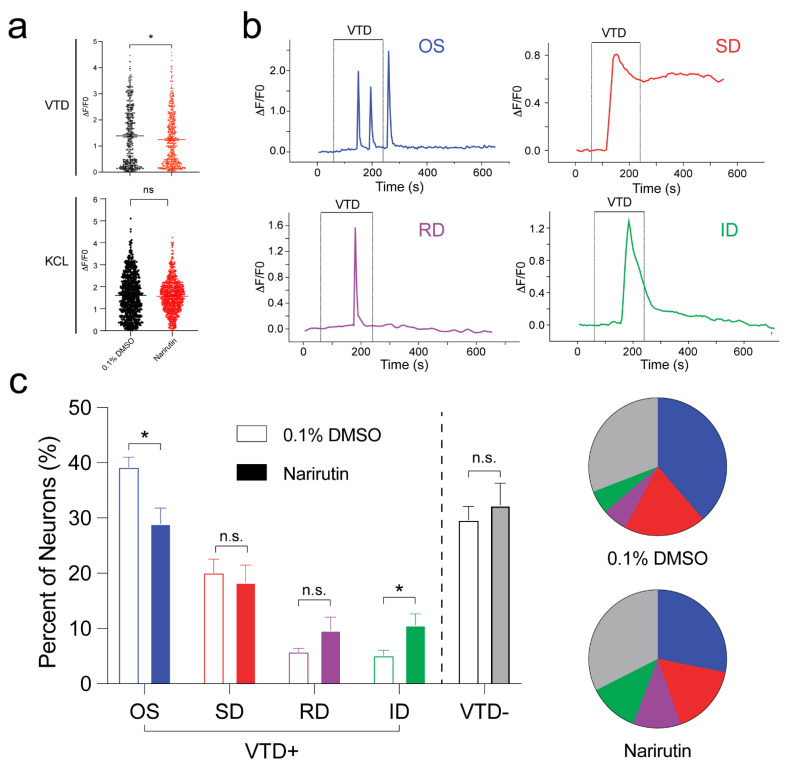
Narirutin inhibited Veratridine-triggered nociceptor activities of DRG neurons. (**a**) Peak Ca^2+^ responses of DRG neurons induced by Veratridine (up) or KCL (down) with 0.1% DMSO or 20 μM Narirutin overnight incubation (0.1% DMSO, *n* = 5 slices; Narirutin, *n* = 4 slices). (**b**) Representative traces of the four response profiles triggered by Veratridine: oscillatory (OS, blue), slow decay (SD, red), rapid decay (RD, purple), and intermediate decay (ID, green) profile. (**c**) Percentage of different populations of DRG neurons based on the Veratridine-response pattern after being treated with 0.1% DMSO or 20 μM Narirutin overnight incubation. Pie charts represent mean values in the histogram (0.1% DMSO, *n* = 5 slices; Narirutin, *n* = 4 slices). Data are expressed as means ± SEM. Asterisks indicate statistical significance compared with vehicle treatment (* *p* < 0.05, unpaired two-tailed Student’s *t*-test and two-way ANOVA with Sidak’s post hoc test).

**Figure 3 ijms-23-14842-f003:**
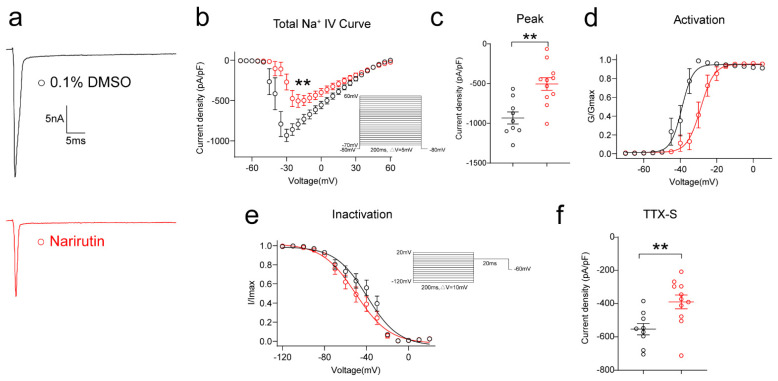
Narirutin reduces the total Na^+^ currents in DRG sensory neurons. (**a**) Representative traces of the total Na^+^ currents from DRG sensory neurons treated with 0.1% DMSO (control) or 20 µM Narirutin. Currents were evoked by 200 ms pulse between −70 and +60 mV. Summary of (**b**) the normalized (pA/pF) sodium current density versus voltage relationship and (**c**) peak total Na^+^ current density at −10 mV from DRG neurons treated as indicated. Boltzmann fits for normalized conductance, G/Gmax voltage relationships for voltage-dependent (**d**) activation, and (**e**) inactivation sensory neurons. The half-maximal activation (V_1/2_) and slope values (*k*) for activation and inactivation are summarized in Table 1. (**f**) Summary of peak TTX sensitivity (TTX-S) Na^+^ current densities for DRG neurons treated as described above. The TTX sensitivity fractions were calculated as described in the Methods section. Data are expressed as means ± SEM. Asterisks indicate statistical significance compared with vehicle treatment (** *p* < 0.01, Mann–Whitney test, *n* > 9 cells per condition).

**Figure 4 ijms-23-14842-f004:**
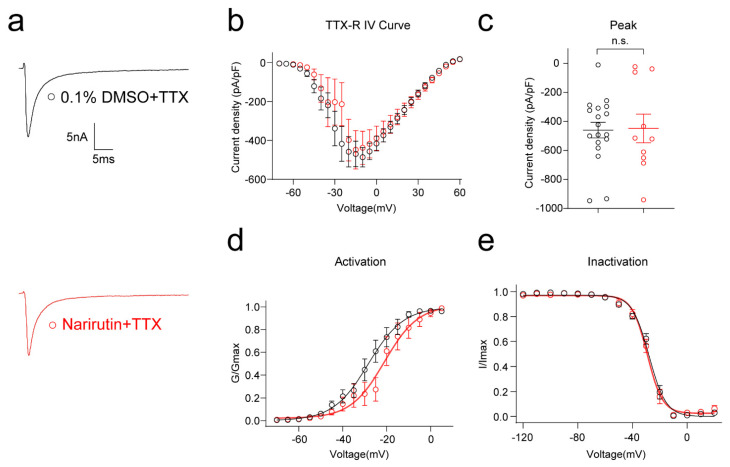
Narirutin did not affect TTX-R currents in DRG sensory neurons. (**a**) Representative traces of the TTX-R currents from DRG sensory neurons treated overnight with 0.1% DMSO + 1 μM TTX or 20 µM Narirutin + 1 μM TTX. Currents were evoked by 200 ms pulse between −70 and +60 mV. Summary of (**b**) the normalized (pA/pF) sodium current density versus voltage relationship and (**c**) peak TTX-R current density at −10 mV from DRG neurons treated as indicated. Boltzmann fits for normalized conductance, G/Gmax voltage relationships for voltage-dependent (**d**) activation, and (**e**) inactivation sensory neurons. The half-maximal activation (V_1/2_) and slope values (*k*) for activation and inactivation are summarized in Table 1. Data are expressed as means ± SEM. (*p* > 0.05, Mann–Whitney test, *n* > 9 cells per condition).

**Figure 5 ijms-23-14842-f005:**
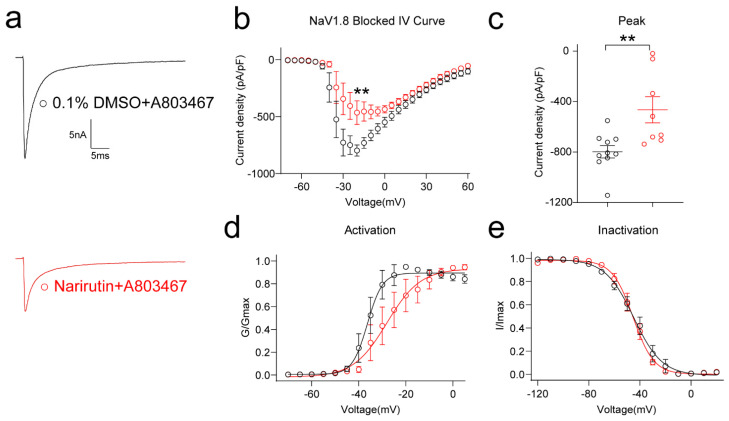
Effect of Narirutin on sodium currents after blocking Na_v_1.8 currents by A-803467. (**a**) Representative traces of the Na^+^ currents from DRG sensory neurons after being treated with 0.1% DMSO + 500 nM A-803467 or 20 µM Narirutin + 500 nM A-803467. Currents were evoked by 200 ms pulse between −70 and +60 mV. Summary of (**b**) the normalized (pA/pF) sodium current density versus voltage relationship and (**c**) peak Na^+^ current density (Na_v_1.8 blocked) at −10 mV from DRG neurons treated as indicated. Boltzmann fits for normalized conductance, G/Gmax voltage relationships for voltage-dependent (**d**) activation, and (**e**) inactivation sensory neurons. The half-maximal activation (V_1/2_) and slope values (*k*) for activation and inactivation are summarized in Table 1. Data are expressed as means ± SEM. Asterisks indicate statistical significance compared with vehicle treatment (** *p* < 0.01, Mann–Whitney test, *n* > 9 cells per condition).

**Figure 6 ijms-23-14842-f006:**
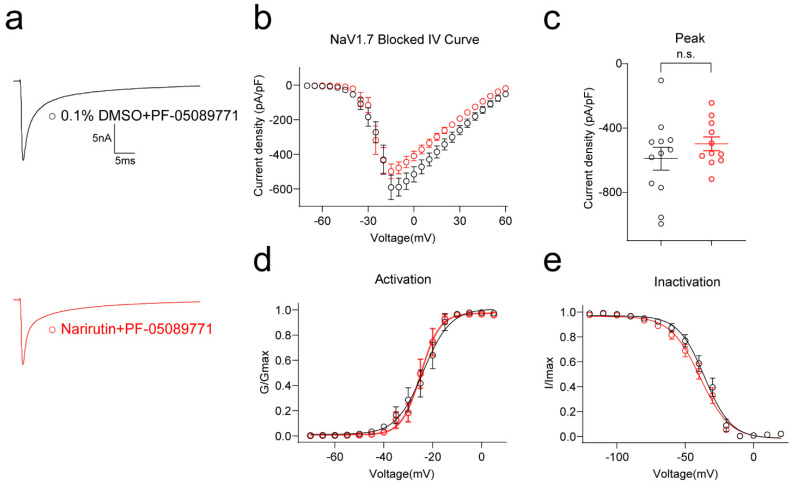
Na_v_1.7 inhibitor PF-05089771 blunted the effect of Narirutin on sodium currents. (**a**) Representative traces of Na^+^ currents in DRG sensory neurons after blocking Na_v_1.7 with 0.1% DMSO + 100 nM PF-05089771 or 20 µM Narirutin + 100 nM PF-05089771 treatment. Currents were evoked by 200 ms pulses between −70 and +60 mV. Summary of (**b**) the normalized (pA/pF) sodium current density versus voltage relationship and (**c**) peak Na^+^ current density (Na_v_1.7 blocked) at −10 mV from DRG neurons treated as indicated. Boltzmann fits for normalized conductance, G/Gmax voltage relationships for voltage-dependent (**d**) activation, and (**e**) inactivation sensory neurons. The half-maximal activation (V_1/2_) and slope values (*k*) for activation and inactivation are summarized in Table 1. Data are expressed as means ± SEM. (*p* > 0.05, Mann–Whitney test, *n* > 9 cells per condition).

**Figure 7 ijms-23-14842-f007:**
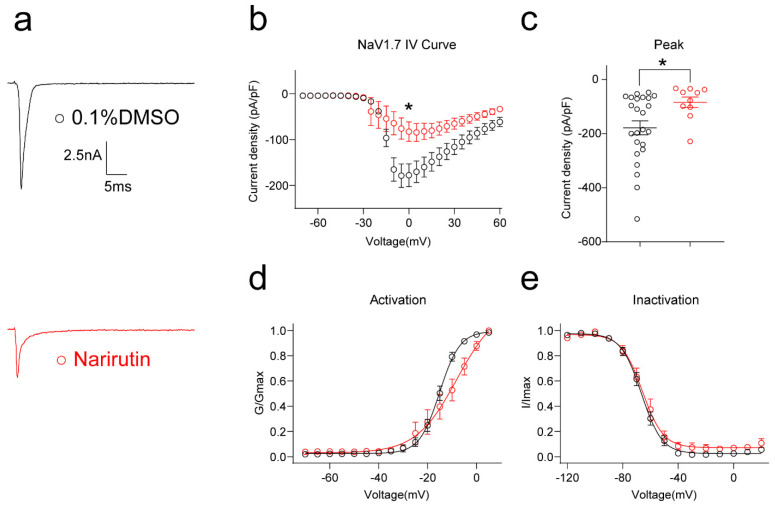
Narirutin reduces the Na_v_1.7 currents in the HEK293 cell line. (**a**) Representative traces of the Na_v_1.7 currents from HEK293 cells transfected using pcDNA3.1- Na_v_1.7-Flag with GFP plasmid and incubated overnight with 0.1% DMSO or Narirutin 20 μM. Currents were evoked by 200 ms pulse between −70 and +60 mV. Summary of (**b**) the normalized (pA/pF) sodium current density versus voltage relationship and (**c**) peak Na_v_1.7 current density at −10 mV from HEK293 cells treated as indicated. Boltzmann fits for normalized conductance, G/Gmax voltage relationships for voltage-dependent (**d**) activation, and (**e**) inactivation sensory neurons. The half-maximal activation (V_1/2_) and slope values (*k*) for activation and inactivation are summarized in Table 1. Data are expressed as means ± SEM. Asterisks indicate statistical significance compared with vehicle treatment (* *p* < 0.05, Mann–Whitney test, *n* > 9 cells per condition).

**Figure 8 ijms-23-14842-f008:**
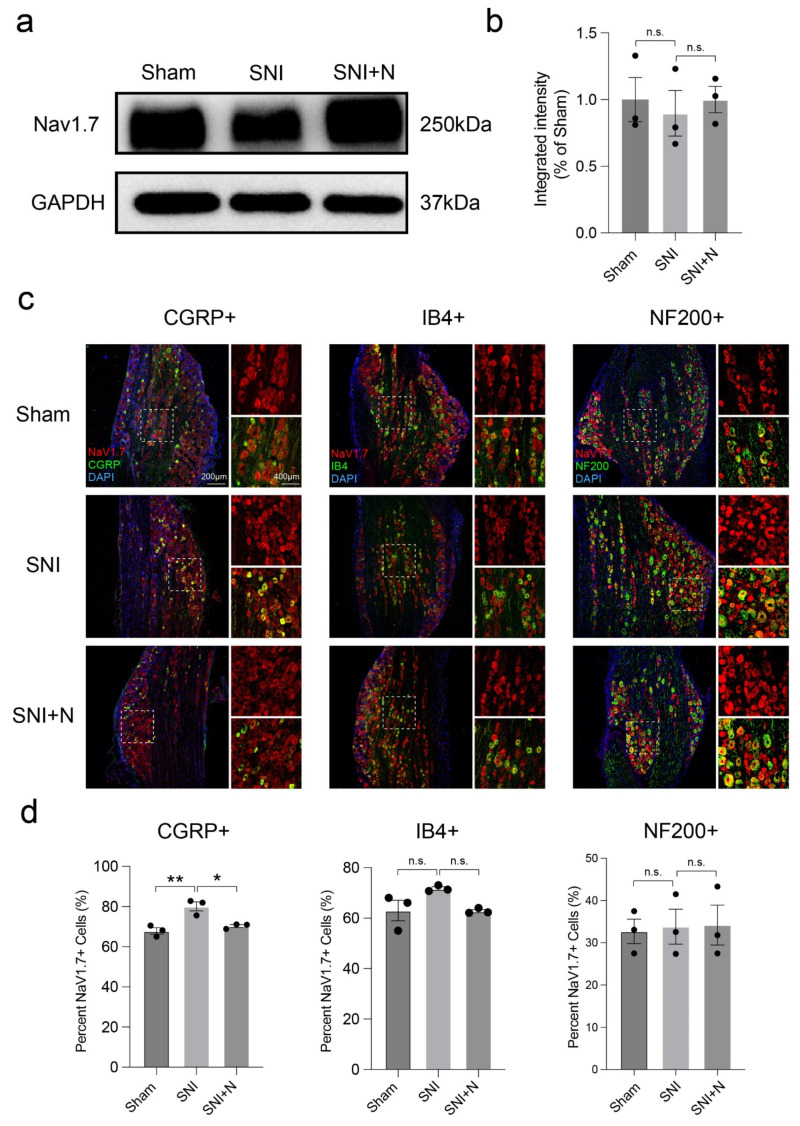
Expression of Na_v_1.7 in different types of DRG sensory neurons. (**a**,**b**) Western blot analysis expression level of Na_v_1.7 isolated from lysates of L4-L6 DRGs from the ipsilateral side of the lumbar spine among Sham, SNI, and SNI treated with Narirutin (SNI + N). (**c**) Representative immunofluorescence images of Na_v_1.7 in DRG neurons of the ipsilateral side of L4-L6 DRGs slices from Sham, SNI, and SNI + N with corresponding co-localization (white arrow). Nuclei were stained with DAPI (blue). Left: peptidergic neurons were labeled using calcitonin gene-related peptide (CGRP, green), which was co-labeled with Na_v_1.7 (red). Merged images show CGRP-containing peptidergic neurons co-labeled with Na_v_1.7 (yellow). Middle: non-peptidergic sensory neurons are labeled using Isolectin B4 (IB4, green), which is co-labeled with Na_v_1.7 (red) and is shown in the merged image (yellow). Right: myelinated neurons were labeled using NF200 (green), which was co-localized with Na_v_1.7 (red), and the merged image shows its co-localization (yellow). (**d**) Co-localization analysis of CGRP, IB4, NF200 with Na_v_1.7 in DRG neurons. Data are expressed as means ± SEM. Asterisks indicate statistical significance compared with vehicle treatment (* *p* < 0.05, ** *p* < 0.01, one-way ANOVA followed by Tukey’s test, *n* = 3 rats per group).

**Table 1 ijms-23-14842-t001:** Effects of Narirutin on Gating Properties of Voltage-gated Channels in DRG Neurons and HEK cell line ^1^.

**Total Na_V_** (**Narirutin-20 μM**)		**0.1% DMSO**	**Narirutin**	***p* Value ^2^**
**Treated**		**Overnight**	**Overnight**	
Activation	V_1/2_	−39.2 ± 0.7	−28.8 ± 0.7	**** *p* < 0.0001
	*k*	3.0 ± 0.6	3.7 ± 0.6	*p* = 0.4585
Inactivation	V_1/2_	−39.5 ± 2.0	−50.3 ± 2.0	** *p* = 0.0013
	*k*	−14.9 ± 1.9	−16.0 ± 2.0	*p* = 0.7018
**Blocked TTX-S** (**TTX-1 μM**)		**0.1% DMSO + TTX**	**Narirutin + TTX**	
**Treated**		**Another Overnight** **TTX Acute**	**Another Overnight** **TTX Acute**	
Activation	V_1/2_	−28.3 ± 1.2	−21.0 ± 1.9	** *p* = 0.0045
	*k*	8.0 ± 1.2	7.7 ± 1.7	*p* = 0.8655
Inactivation	V_1/2_	−27.8 ± 0.5	−29.1 ± 0.6	*p* = 0.1025
	*k*	−6.2 ± 0.4	−5.6 ± 0.5	*p* = 0.3450
**Blocked Na_v_1.8** (**A-803467-500 nM**)		**0.1%DMSO + A-803467**	**Narirutin + A-803467**	
**Treated**		**Another Overnight** **A-803467 Acute**	**Another Overnight** **A-803467 Acute**	
Activation	V_1/2_	−36.3 ± 0.8	−28.0 ± 1.9	** *p* = 0.0026
	*k*	3.2 ± 0.7	7.1 ± 1.8	*p* = 0.0681
Inactivation	V_1/2_	−44.9 ± 1.2	−45.3 ± 0.8	*p* = 0.7653
	*k*	−11.1 ± 1.1	−8.1 ± 0.7	* *p* = 0.0412
**Blocked Na_v_1.7** (**PF-05089771-100 nM**)		**0.1%DMSO + PF-05089771**	**Narirutin + PF-05089771**	
**Treated**		**Another Overnight** **PF-05089771 Acute**	**Another Overnight** **PF-05089771 Acute**	
Activation	V_1/2_	−23.8 ± 0.9	−24.9 ± 0.6	*p* = 0.3620
	*k*	5.5 ± 0.8	3.9 ± 0.6	*p* = 0.1084
Inactivation	V_1/2_	−35.8 ± 1.1	−39.9 ± 1.0	* *p* = 0.0398
	*k*	−9.6 ± 0.9	−10.8 ± 0.9	*p* = 0.3740
**Na_v_1.7** (**Narirutin 20 μM**)		**0.1% DMSO**	**Narirutin** (**20 μM**)	
**Treated**		**Overnight**	**Overnight**	
Activation	V_1/2_	−15.2 ± 0.4	−8.0 ± 2.9	* *p* = 0.0342
	*k*	4.1 ± 0.3	8.3 ± 1.6	* *p* = 0.0295
Inactivation	V_1/2_	−66.5 ± 0.6	−65.8 ± 1.1	*p* = 0.5900
	*k*	−7.4 ± 0.6	−7.5 ± 0.9	*p* = 0.9140

^1^ Values are means ± SEM calculated from fits of the data from the indicated number of individual cells to the Boltzmann equation; V_1/2_, midpoint potential (mV) for voltage-dependent activation or inactivation; *k*, slope factor. ^2^ Significantly different from the value for DMSO (* *p* < 0.05, ** *p* < 0.01, **** *p* < 0.0001, Student’s *t*-test).

## Data Availability

Not applicable.

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
