# Peer review of "Reversal of Peripheral Neuropathic Pain by the Small-Molecule Natural Product Narirutin via Block of Nav1.7 Voltage-Gated Sodium Channel"

_ijms, 2022, doi:10.3390/ijms232314842_

Round 1
Reviewer 1 Report
The results of the study are relevant to improve the understanding of the physiological and pathophysiological mechanisms related to alterations in the nociceptive pathway and provide valuable information for possible pharmacological or pharmacotherapeutic alternatives for its modulation, such as the indicated by the authors: narirutin. From the point of view of experimental design, it is a well-structured work with the appropriate techniques according to the objectives pursued.
The findings on narirutin derived from Citrus unshiu on voltage-dependent sodium channels present coherent results in a good formulation of the structure of the work, However, the conclusions regarding its antinociceptive effects should be corrected, since although it is most likely that the nociceptive pathway is modified through its effects on voltage-dependent sodium channels or changes in nociceptor populations (such as OS), they are not strictly valid to indicate an antinociceptive property, since to comment or conclude on them, the corresponding behavioural experiments with naruritin should be carried out or bibliographically substantiated naruritin nociceptive effect must be included in order to discuss them in more detail.
The reference to the method used to obtain the neuropathic pain model must be revised and well referenced. Reference 36 does not contain any description of Spared Nerve injury (SIN).
In addition, it is more advisable to discuss the effects on nociception rather than pain, being the latter a perception of damage with multiple subjective factors that are far from the assessments presented in this paper.
Specific comments,
Line 3, 25, 28, 29 etc: The authors due follow the IUPHAR nomenclature of sodium receptors
Line 26: The authors should Specified the rodent specie of DRG and, if applicable, indicate the place of extraction of the sample (e.g. L4-L6).
Line 38: The author describe in some parts of the work in terms od efficiency but it is advisable to discuss in terms of efficacy
Line 55: the name of the tarantula should be put in italics.
Line 73: only the term in vitro would be the most appropriate for these experiments in DRG and HEK cells.
Figure 1: Scatter Plot is recommended for representation in C and the range for measuring this parameter should be indicated.
Line 111: Correct and standardize Na+ nomenclature
Line 114, 115, etc: Authors should standardize the nomenclature to use Ca2+ or calcium
Why in section 2.2 the authors use only DMSO as a vehicle while in section 2.1 a different vehicle (10% DMSO, 10% Tween-80, 80% Saline, 20μL) is used.
Line 132: Correct slides
Line 133-135: The use of upper/lower case letters should be homogenized; for example: Veratridine/veratridine; narirutin/Narirutin etc
Line 151: Correct the peak current density units along the work. It must be used pA/pF
Line 163: V1/2 in italics?
Figure 3. The scale used in a does not correspond to that described in the text and figure caption (200 ms).
Table 1. Coorect the TTX concentration units for TTX
Line 182: Correct the limits for statistical significance
Line 187: The abbreviation for TTX-S must be placed in previous paragraphs.
Line 213: Correct “corelated”
Figure 5: Please the authors should correct the complete Figure 5. Symbols and numbers appear that do not correspond to the work.
Figure 8: It is recommended to improve the quality of the whole image, especially the immunofluorescence images.
Author Response
General comments:
- The findings on narirutin derived from Citrus unshiu on voltage-dependent sodium channels present coherent results in a good formulation of the structure of the work, However, the conclusions regarding its antinociceptive effects should be corrected, since although it is most likely that the nociceptive pathway is modified through its effects on voltage-dependent sodium channels or changes in nociceptor populations (such as OS), they are not strictly valid to indicate an antinociceptive property, since to comment or conclude on them, the corresponding behavioural experiments with naruritin should be carried out or bibliographically substantiated naruritin nociceptive effect must be included in order to discuss them in more detail.
Reply: Thank you very much for your suggestions. Behavioral experiment is one of the best ways to evaluate the antinociceptive property of Narirutin. In our manuscript, we have assessed the antinociceptive ability of Narirutin on mechanical allodynia by Von Frey test (Figure 1b, c), a classic behavioral experiment to evaluate rodent’s mechanical hypersensitivity. In order to add reliability, we have supplemented acetone test for SNI rodents treated with Narirutin or vehicle, a behavioral test that evaluates rodent’s cold hypersensitivity. As a result, both Von Frey test and acetone test have strengthened an antinociceptive property of Narirutin. Together with the whole-cell patch clamp and molecular experiments, we can conclude that Narirutin is an efficient analgesic in alleviating neuropathic pain in rodent models of spared nerve injury.
- The reference to the method used to obtain the neuropathic pain model must be revised and well referenced. Reference 36 does not contain any description of Spared Nerve injury (SIN).
Reply: We have modified the literature referenced in constructing the SNI pain model.
- In addition, it is more advisable to discuss the effects on nociception rather than pain, being the latter a perception of damage with multiple subjective factors that are far from the assessments presented in this paper.
Reply: Thank you for the suggestion. In this study, we evaluated the analgesic effect of Narirutin on SNI model, which is a classic pre-clinical rodent model mimicking neuropathic pain symptoms. We strengthened the link between Narirutin and neuropathic pain by performing acetone test. Thus, together with Von Frey test, we concluded that Narirutin relieved SNI-induced neuropathic pain.
Specific comments:
- Line 3, 25, 28, 29 etc: The authors due follow the IUPHAR nomenclature of sodium receptors
Reply: We have revised the previous irregularities in the naming of sodium channels according to the IUPHAR nomenclature (e.g. Nav1.7 or Nav).
- Line 26: The authors should Specified the rodent specie of DRG and, if applicable, indicate the place of extraction of the sample (eg. L4-L6).
Reply: We have clarified the origin and species of DRGs (Rat, L4-L6).
- Line 38: The author describe in some parts of the work in terms od efficiency but it is advisable to discuss in terms of efficacy
Reply: Thank you for your suggestion, we have revised the relevant content in this place (efficiency to efficacy).
- Line 55: the name of the tarantula should be put in italics.
Reply: Based on your suggestion, we have standardized the rules for writing them (tarantula to tarantula).
- Line 73: only the term in vitro would be the most appropriate for these experiments in DRG and HEK cells.
Reply: According to the problem you have indicated, we have modified it to “in vitro experiments”.
- Figure 1: Scatter Plot is recommended for representation in C and the range for measuring this parameter should be indicated.
Reply: Based on your suggestion, we have changed the way the diagram is presented from box plot to scatter plot.
- Line 111: Correct and standardize Na+ nomenclature
Reply: Based on the issue you pointed out, we have corrected it to NaV.
- Line 114, 115, etc: Authors should standardize the nomenclature to use Ca2+ or calcium
Reply: In light of your comment, we have changed them all to Ca2+.
- Why in section 2.2 the authors use only DMSO as a vehicle while in section 2.1 a different vehicle (10% DMSO, 10% Tween-80, 80% Saline, 20μL) is used.
Reply: Thank you very much for your advice. High DMSO concentration may lead to cytotoxicity and side effects[1]. Since the limitation for DMSO concentration in in vivo and in vitro experiments varies, we used different dissolution methods of Narirutin in the two conditions. In in vitro experiments, we limited the DMSO concentration to 0.1%, which allowed us to get the final narirutin concentration to 20μM on cells. In in vivo experiments, however, we used different dissolution methods due to the solubility of Narirutin (20μg in 20μL per rat) and the limitation of DMSO concentration (≤10%)[2]. These differences will not change the reliability of the experimental results.
- Line 132: Correct slides
Reply: Based on your comment, we have changed them all to “slices”.
- Line 133-135: The use of upper/lower case letters should be homogenized; for example: Veratridine/veratridine; narirutin/Narirutin etc
Reply: According to the problem you have indicated, we have corrected them (for example, Veratridine or Narirutin).
- Line 151: Correct the peak current density units along the work. It must be used pA/pF
Reply: Based on the issue you pointed out, we have corrected the units of current density to pA/pF.
- Line 163: V1/2 in italics?
Reply: Based on the problem you pointed out, we removed the italics from V1/2.
- Figure 3. The scale used in a does not correspond to that described in the text and figure caption (200 ms).
Reply: Based on the issues you have identified, we have changed the scale and presentation of the current representative curves to match the description of the voltage protocol used for whole-cell recording.
- Table 1. Correct the TTX concentration units for TTX
Reply: According to the problem you pointed out, we changed the TTX concentration unit to “μM”.
- Line 182: Correct the limits for statistical significance
Reply: According to the problem you have indicated, we have revised it to *p < 0.05, **p<0.01, ****p< 0.0001.
- Line 187: The abbreviation for TTX-S must be placed in previous paragraphs.
Reply: We have placed the acronym TTX-S where it first appears (Figure 3, f).
- Line 213: Correct “corelated”
Reply: We have revised it based on the issues you have pointed out (correlated, at line 343).
- Figure 5: Please the authors should correct the complete Figure 5. Symbols and numbers appear that do not correspond to the work.
Reply: We have corrected all mistakes in Figure 5.
- Figure 8: It is recommended to improve the quality of the whole image, especially the immunofluorescence images.
Reply: We have improved the quality of Figure 8.

Reviewer 2 Report
The paper entitled “Reversal of Peripheral Neuropathic Pain by the Small-Molecule Natural Product Narirutin via Block of Nav1.7 Voltage-Gated Sodium Channel” by Yang and coll., investigates the functional effects prompted by the natural product naritutin on NaV channels and pain sensitivity by using a combined approach, including electrophysiology, immunofluoresce, western-blotting experiments, performed in both clonal cells and animal models.
They found that this compound is able to block Nav1.7 channels and shows analgesic properties in a mous model of neuropathic pain.
Some of the results obtained are clearly explained and discussed; however, in some cases additional experiments would clarify the results obtained, thus improving the quality and the impact of the paper.
Major comments:
1. The experiments in Figure 2 appear overinterpreted: in fact, the Authors described narirutin-induced effects on calcium transients in distinc classes of DRG neurons. These effects could be a consequence of NaV channel block; however, same effects could also be explained by narirutin-induced direct blockade of Cav channels. The Authors should perform additional experiments allowing to exclude this possibility.
2. The results reported in Table 1 are not well explained and failed to report important information. In particular, the biophysical values (for activation and inactivation) of currents recorded after o.n. narirutin incubation are not reported for each experimental condition, thus rendering more difficult the interpretation of the results (in particular, it’s not always clear which groups should be compared). As an example, the results obtained after TTX treatment are not clear. This paragraph should be rephrased.
3. The Authors should mention how they choose specific drug concentrations (as an example for A-803467).
4. Na+ currents are not clearly solved: the reported traces appear to be acquired at a too low frequency: the Authors should improve recording modalities (or traces presentation) in order to make more visible differences between groups in terms of Na+ currents.
5. The results shown in figure 8 should be improved: in particular, in the case of the western blot image, the Authors should use a positive and negative control (as an example, non transfected or transfected HEK cells with the Nav1.7 plasmid), to demonstrate the specificity of the band recognized by the antibodies used. In addition, the images used for colocalization experiments should be improved in clarity.
Minor comments:
1. Line 197, the expression “voltage-contingent activation” is not clear.
2. In some cases, results are expressed in a not explained manner. As an example, at lanes 191-192 it is reported “(Figure 3f, 0.1% DMSO, -553.9±34.0 mV (n=9); narirutin, -389.7±41.9 mV (n=10), p=0.0097; 191 Mann-Whitney test)”: please, explain which “mV” is referred to.
3. In Figure 5 there are problems with some figure legends, that appear non readable.
Author Response
Major comments:
- The experiments in Figure 2 appear overinterpreted: in fact, the Authors described narirutin-induced effects on calcium transients in distinc classes of DRG neurons. These effects could be a consequence of NaV channel block; however, same effects could also be explained by narirutin-induced direct blockade of Cav channels. The Authors should perform additional experiments allowing to exclude this possibility.
Reply: Thank you for the suggestion. To address whether Cav channels were involved, we used 40mM KCl to trigger DRG neurons and found narirutin didn’t change the calcium response of DRGs compared with the control group, this result indicated the effect of narirutin on veratridine triggered calcium transients didn’t involve Cav channels, as shown in Figure 2b.
- The results reported in Table 1 are not well explained and failed to report important information. In particular, the biophysical values (for activation and inactivation) of currents recorded after o.n. narirutin incubation are not reported for each experimental condition, thus rendering more difficult the interpretation of the results (in particular, it’s not always clear which groups should be compared). As an example, the results obtained after TTX treatment are not clear. This paragraph should be rephrased.
Reply: In response to your question about Table 1, we have redesigned the table to meet the requirements. We have mainly clarified the source of the data related to each section with the corresponding title, and clarified the corresponding blocking agent or drug delivery method. See revised Table 1 for details.
- The Authors should mention how they choose specific drug concentrations (as an example for A-803467).
Reply: Our selection of relevant drug concentrations is based mainly on previous journals published by us or other professional researchers in relevant fields. For example, TTX-1μM [3], A-803467-500nM [4], PF-05089771-100nM [5].
- Na+ currents are not clearly solved: the reported traces appear to be acquired at a too low frequency: the Authors should improve recording modalities (or traces presentation) in order to make more visible differences between groups in terms of Na+ currents.
Reply: The signals were sampled at a rate of 20 kHz, which is fast enough for recording Na+ currents. Besides, we have changed the presentation of the representative current curves according to the suggestion.
- The results shown in figure 8 should be improved: in particular, in the case of the western blot image, the Authors should use a positive and negative control (as an example, non transfected or transfected HEK cells with the Nav1.7 plasmid), to demonstrate the specificity of the band recognized by the antibodies used. In addition, the images used for colocalization experiments should be improved in clarity.
Reply: We performed positive control validation of Nav1.7 antibody specificity using HEK 293 cells transfected with Nav1.7 plasmid or vector, details of the data are shown in Figure S1 below. The Nav1.7 antibody can specifically recognize the expression of Nav1.7 in both groups, while the expression level of Nav1.7 in HEK293 cells transfected with Nav1.7 plasmid is significantly higher than those treated with vector. Also, we have improved the quality of Figure 8.
Figure S1. Expression of Nav1.7 in HEK 293 cells. (a) Representative blots and (b) analysis of the expression level of Nav1.7 isolated from HEK293 cells transfected with Nav1.7 plasmid and vector. Data are expressed as means ± SEM. Asterisks indicate statistical significance compared with vehicle treatment (*p<0.05, two-tailed unpaired student’s t test, n=3 rats per group).
Minor comments:
- Line 197, the expression “voltage-contingent activation” is not clear.
Reply: We have modified “voltage-contingent activation” to “the voltage-dependent activation and inactivation properties of TTX-R Na+ channels in DRG neurons were affected by Narirutin”.
- In some cases, results are expressed in a not explained manner. As an example, at lanes 191-192 it is reported “(Figure 3f, 0.1% DMSO, -553.9±34.0 mV (n=9); narirutin, -389.7±41.9 mV (n=10), p=0.0097; 191 Mann-Whitney test)”: please, explain which “mV” is referred to.
Reply: Our interpretation of the results consists of two parts in the manuscript. First, we summarize the results in general language, followed by the results' details. We believe that this form can better explain our experimental results. Besides, we have modified it here by specifying the units after each corresponding data and changing “mV” to “pA/pF”.
- In Figure 5 there are problems with some figure legends, that appear non readable.
Reply: We have corrected all mistakes in Figure 5.

Round 2
Reviewer 2 Report
No additional comments.